



# Full-Scale Wind Turbine Performance Assessment: A Study of Aerodynamic Degradation and Operational Influences

Tahir H. Malik[1] and Christian Bak[2]

[1]Vattenfall, Amerigo-Vespucci-Platz 2, 20457, Hamburg, Germany
[2]DTU Wind and Energy Systems, Frederiksborgvej 399, 4000 Roskilde, Denmark
**Correspondence:** Tahir H. Malik (tahir.malik@vattenfall.de)

**Abstract.** This study investigates how blade aerodynamic modifications, including Leading Edge Roughness (LER), influence offshore wind turbine performance over their operational lifespan. Developing a novel methodology, this research analyses data from twelve multi-megawatt turbines over a twelve-year period, focusing on the intricate relationship between blade erosion, blade enhancements, operations and maintenance events, control PLC parameter updates, and their cumulative impact on turbine efficiency. The analysis hinges on the integration of SCADA data, Operations and Maintenance (O&M) records, and air density corrections. A key contribution is the development of a Turbine Performance Integral (TPI) method, which leverages generator speed and power output data to track performance trajectories. Seasonal-Trend decomposition using Locally Estimated Scatterplot Smoothing (STL) further isolates long-term trends and seasonal variations in performance. Overcoming data availability and quality limitations, the study reveals significant findings concerning software updates impacts on turbine control strategies, the variable effects of blade repairs and enhancements and the complex interaction between O&M events and performance. This study's strength lies in its methodical approach and statistical rigour, offering a path forward in the quest for optimised wind turbine efficiency and advancing renewable energy.

## 1 Introduction

The detrimental effect of Leading Edge Roughness (LER) or Leading Edge Erosion (LEE) on aerofoil characteristics has been investigated through wind tunnel experiments and various studies on the impact of erosion and roughness on wind turbine annual energy production (AEP) Mishnaevsky Jr et al. (2021). Predictions suggest that erosion related annual energy losses of up to 7% may occur described in various publications Han et al. (2018) Maniaci et al. (2016) Bak et al. (2020) Bak (2022). However, a key challenge remains in identifying and validating these computed energy losses when analysing SCADA data from operational wind turbines Ding et al. (2022). This challenge is manifested by a continued absence of an established correlation between blade erosion and turbine performance underscoring a lack of deeper understanding of the underlying reasons.

Wind turbine performance assessment is especially challenging in the academic and owner-operator environment due to limited access to quality data from wind turbines Pandit et al. (2023) Leahy et al. (2019). Where data is accessible, it is to a restricted set of sensors or SCADA channels, typically sampled at a lower rate, or it may be available for very limited



time periods Yang et al. (2014). Such restrictions hinder comprehensive data analysis and conclusive research findings Badihi et al. (2022) Gonzalez et al. (2019). Usually, little to no information on specifications of the turbine, such as blade profiles or aerofoil polars is available such as those tested by Krog Kruse et al. (2021) Gaudern (2014), with several reasons contributing to this limitation. One of the primary reasons being intellectual property protection by Original Equipment Manufacturers (OEMs), since their significant research and development (R&D) investments help them maintain their advantage in a highly

competitive market. Additionally, suppliers are often hesitant to empower customers with detailed data that could be leveraged in, for example, a performance warranty claim. Moreover, the owner-operator faction of the industry has seen limited success in gleaning valuable turbine performance data from SCADA systems Ding et al. (2022), which has not created a compelling incentive for a push towards suppliers to provide more comprehensive data, nor has it sparked a cost-benefit argument for owner-operators for investing in higher quality data or more expensive dedicated CMS systems Yang et al. (2014) Tautz-Weinert

and Watson (2017). This situation has culminated in a lack of drive from customers to push for better data accessibility and quality, exacerbating the challenge of wind turbine performance assessment as well as impeding innovation and progress in the field Pandit et al. (2023). Furthermore, the controller of the wind turbine often remains a 'black box' Aho et al. (2012) to owners, operators and academics, making the assessment of the turbine's behaviour in response to atmospheric conditions, O&M events, upgrades or blade conditions highly challenging, leading to inconclusive or, in the worst case, incorrect conclusions.

Developing a reliable performance evaluation method is crucial as is enhancing the efficiency of current wind turbine fleets, where even marginal performance gains can translate into significant economic and environmental benefits.

To address these challenges, wind turbine SCADA data was analysed and an understanding of the wind turbine controller underlying turbine behaviour in response to atmospheric conditions, O&M events and modified blade aerodynamics was applied. The goal was to discern the effect of leading edge roughness from the numerous variables and uncertainties affecting

its performance, which may be likened to finding a needle in a haystack. The developed method avoids the need for direct wind speed measurements, reducing uncertainty and providing insights previously difficult to achieve with currently available SCADA data Butler et al. (2013); Albers (2012).

This study aims to dissect measurement data from twelve multi-megawatt offshore wind turbines to determine the specific effects of modified blade aerodynamics on their performance and separate these relatively minor effects from the plethora of

events that affect the wind turbine over its lifetime.

The distinguishing aspect of this work lies in the development of a method where a single turbine indicates performance shifts, without use of wind speed measurements, customised for the OEM and the turbine's specific controller. Comprehensive operations and maintenance (O&M) data for wind turbines is a rarity in the academic field, as is the opportunity to modify blade aerodynamics to assess its impact on turbine performance. The present investigation leverages these aspects to assess turbine

performance, creating insights to enable more informed business decisions, predictive maintenance strategies and ultimately enhance the energy yield as well as longevity of the turbines. Ultimately, shedding a light on how many kilowatt hours are lost due to the limited understanding of field performance influenced by blade erosion.





## 2 Method

The present study employs a detailed analysis of wind turbine measurement SCADA data to assess the influence on performance due to various factors. These causes include seasonal variations, O&M events and turbine component degradation. A particular emphasis is laid on evaluation of effect on performance due to modified blade aerodynamics due to blade erosion and blade enhancements. These particular causes of performance alterations are known to have relatively smaller performance deviations, especially when compared to other, more prominent sources of variation such as main component malfunctions Dao et al. (2019) (find better REFERENCE) Yang et al. (2014). The presented methodology, concentrates on the separation of causes of variation of performance deviations and their individual contributions. For this purpose twelve front-row, offshore multi-megawatt turbines within a wind farm were selected for their direct exposure to undisturbed dominant wind conditions. Due to confidentiality reasons neither the site nor the wind turbine type shall be described in detail.

### 2.1 Data Collection

#### 2.1.1 Operations and Maintenance (O&M) Data

A detailed compilation of maintenance records was undertaken, which were subsequently consolidated with the SCADA data. This dataset included dates and details for inspection and maintenance activities such as blade repair, blade leading edge protection (LEP), application of blade enhancements such as aftermarket vortex generators and Gurney flaps, component repairs or replacements, turbine curtailments and outages etc. An auxiliary source of data was a System Applications and Products (SAP) or accounting database, that held records of O&M billing dates and respective costs . However, due to the transactional nature of the SAP records, they were used only to a very limited extent as their date did not reflect the actual date an O&M event took place on a wind turbine, rather the date when the work was billed to the business.

It is important to recognise the challenges in assembling a complete and precise record of all repairs for the wind farm in question. This can be attributed to occasional gaps in reports, variability in maintenance records over some periods and instances of data loss - similar challenges have been encountered by others in the field Leahy et al. (2019). Recognising repercussions of these discrepancies on the findings shall be addressed and deliberated in Section 3, as the inherent integrity of these records had implications on the depth and accuracy of the analysis.

The O&M events compiled for the selected wind farm were categorised under the following main categories, with a detailed breakdown shown in Figure 8:

– Control PLC parameter Update - Only the approximate date of the last update is recorded. Previous updates' dates are unavailable

– Mechanical Repairs (e.g., gearbox replaced, generator bearing replaced)

– Blade Maintenance (e.g., blade repaired, blade enhancements)

– Sensor and Instrumentation (e.g., wind sensor replaced, generator high temperature)





    – Fluid Maintenance (e.g., gearbox oil change, hydraulic oil change)

– Other

### 2.1.2   Wind Turbine SCADA data

The data used in this study were sourced from twelve individual wind turbines situated in the first row of an offshore wind farm with exposure to predominant wind conditions. This selection mitigated wake effects of other turbines or wind parks. A comprehensive twelve year dataset, with SCADA data sampled every second, was employed to improve accuracy Badihi et al.

(2022) while searching for small performances changes. This of course created computational challenges due to the associated increased volume of data Pandit et al. (2023).

    The SCADA system records various parameters at regular intervals. From the limited available sensors, the following parameters relevant to this study were collected:

    – Nacelle wind speed $\nu$ (m/s)

– Nacelle direction ($^o$)

    – Ambient Temperature T (K)

    – Blade pitch angle $\beta(^o)$

    – Rotor speed $\omega$ (RPM)

    – Generator speed $\Omega$ (RPM)

– Power production P (kW)

    – Power setpoint demand P (kW)

    – Turbine operational state (e.g. waiting for wind, curtailed, cable unwind)

    Additionally, relevant parameters were collected from an onsite meteorological mast (located on the offshore substation) and the EMD International database (via WindPRO EMD International A/S (2023)). The following obtained metrics were used

separately to account and correct for variations in air density, the importance of which was highlighted by Farkas (2011):

    – Atmospheric Pressure P (Pa)

    – Ambient Temperature T (K)

    – Air Density $\rho$ (kg/m$^3$)



## 2.2 Pre-processing

The SCADA dataset utilised in this study was pre-computed from the wind turbine's high-frequency data archive, where a sensor's signal is only updated when a change is recorded. Due to the substantial volume of data generated, it is worth noting that these high frequency records are retained on the turbine for only a limited duration. The employed dataset was one-second interval sampled. Consequently, any encountered missing values were filled using the 'previous value' method, which replaces each missing value with the most recent, preceding, non-missing value. In the absence of historical high-frequency data, this approach to handling missing values reduces the computational demands while attempting to minimise the introduction of potential bias or inaccuracies.

Certain derived variables were also computed to aid data filtering. For instance, the tip speed ratio ($\lambda$) was calculated using rotor speed and horizontal wind speed measurements. Additionally, the nacelle direction is used as a proxy for wind direction despite it being influenced by the turbine's control algorithm hysteresis and rotor wake.

Furthermore, the air density ($\rho$) calculated at the substation was compared with that from the EMD International's database and found to be a close match. Due to occasional gaps in the meteorological mast data, the EMD data, available at hourly intervals, were employed for analysis - of course adding some degree of uncertainty. The gaps between the hourly timestamps were filled using linear interpolation. An adjusted power value was computed based on the obtained air density and was used in all subsequent analysis.

The data was filtered and processed in accordance with the guidelines outlined in International Electrotechnical Commission (IEC) 61400-12-1 Commission et al. (2017).

## 2.3 Density Correction

To isolate wind turbine performance variation influencing factors, it is crucial to adjust for variables that introduce uncertainty, to whatever extent possible. Air density is one such variable Farkas (2011) in the "haystack" of uncertainties that confound the data. Wind kinetic energy being proportional to turbine power, correcting for air density variations over time is essential, as demonstrated by Butler et al. (2013).

The power generated by a wind turbine may be expressed by:

$$P = \frac{1}{2}\rho v^3 A C_p \tag{1}$$

where:

- $P$ is the mechanical power generated by the wind turbine,

- $\rho$ is the air density,

- $A$ is the swept area of the wind turbine (calculated as $\pi\frac{D^2}{4}$ where $D$ is the rotor diameter),

- $v$ is the wind speed, and



- $C_p$ is the power coefficient of a turbine at the design load case. The maximum theoretical value of $C_p$ is 0.593, according to Betz's law.

In this analysis, the turbine power output is adjusted for site-specific temperature, pressure and consequent air density variations. Thus standardising the data and partially reducing an aspect of seasonal or atmospheric induced variations over time. The adjusted power $P_{\text{adj}}$ is calculated by correcting the measured power output $P_{\text{meas}}$ for air density variations. This is achieved by using the density ratio, which is the quotient of the mean air density $\bar{\rho}$ over the instantaneous air density $\rho$ at the time of measurement. The adjusted power is given by:

$$P_{\text{adj}} = P_{\text{meas}} \times \left( \frac{\bar{\rho}}{\rho} \right) \tag{2}$$

where:

- $P_{\text{adj}}$ is the adjusted power output,

- $P_{\text{meas}}$ is the measured power output,

- $\bar{\rho}$ is the mean air density over the measurement period, and

- $\rho$ is the instantaneous air density at the time of measurement.

The mean air density is calculated from the available air density data, and the instantaneous air density is obtained from synchronised environmental obtained from EMD International A/S (2023).

## 2.4 Wind turbine control - Data Analysis

Developing a method for measuring power performance from SCADA data necessitates a comprehensive understanding of the individual turbine and manufacture's control algorithm. Every OEM has developed its unique control strategy to manage turbine loads and optimise power production. These proprietary control philosophies, that are closely guarded intellectual property, highlight their significance in the competitive landscape Aho et al. (2012). Importantly, turbines within the same family can exhibit variations in control strategies due to incremental revisions. Moreover, a turbine's control system may undergo significant changes during its operational lifespan, either through software or hardware upgrades. These upgrades, at times offered as paid customer services, can markedly alter the turbine's performance and the dynamics of performance measurement; this has been ascertained by empirical observations made during this study.

In this context, the chosen approach is tailored to the unique control strategies employed in the particular turbine model under investigation. The specific turbine model utilises both generator torque and blade pitch as primary handles to regulate turbine speed. For the turbines analysed in this study, the blade pitch angle and rotor speed are dynamically adjusted based on the current wind conditions, aiming to optimise energy capture. Consequently, under varying wind conditions, the turbine's generator speed and power output are expected to change dynamically in partial load. Existing literature provides examples





of wind turbine controllers incorporating wind measurement alongside torque measurement Bolik (2004). In this detailed description of the controller it is described how the rotor is operated at low wind speed. And the controller ensuring a mix

of torque as a function of rotor speed and constant tip speed ratio. In this region of the power curve, the blade pitch curve is relatively flat (seen later in Figure 1). At higher wind speeds the rated power is kept constant by pitching the blades.

## 2.5   "Generator Speed-Power Correlation: A Method for Assessing Wind Turbine Performance"

This study introduces an innovative approach to assess wind turbine performance. Traditional power curve analysis (power vs. wind speed) suffers from significant uncertainty in representative wind speed measurement. This work explores an alternative

metric to enhance assessment accuracy. Monitoring the generator speed and power output in the partial load region, marked by its relatively inactive pitch, provides insight into whether the turbine is operating as expected. A discrepancy, where the power output does not increase as expected with increasing generator speed, may indicate potential issues such as mechanical wear and tear, an O&M event, or component inefficiency. Essentially, analysing the relationship generator speed versus power output, in this operational region, can serve as a proxy for turbine performance and a tool or 'virtual sensor' for long-term performance

monitoring. This approach may reveal performance changes due to aspects such as changes in blade aerodynamics due to erosion or application of blade enhancements.

Datasets of each turbine are loaded for the twelve year period. The integral or area under the generator speed curve between 20% and 37% of normalised power is monitored - a region where the pitch angle of the blade is less active (Figure 1). Within the specified power range, an increase in area under the curve, indicates that the turbine has to spin at higher speeds to generate

the same amount of power. Thus, an increase in this area reflects a decrease in the turbine's efficiency. Over the observation period, the trend of this integral can provides a metric for the turbine's performance trajectory: an increasing area suggests a deteriorating performance, while decreasing area indicates an improvement. Thus, the integral, with units of RPM·kW (rad·W/s), is indicative of the turbine's performance trajectory and shall be referred to as Turbine Performance Integral (TPI).

The analysis centres on the use of a ring buffer, which is initialised and then updated weekly to record average values for

each power pin. For the trapezoidal integration calculation within defined bounds, the approach computes the integral under the curve. A consistent buffer size ensures standardised data smoothing across all turbines.

Data processing involves several steps: filling missing bins, updating the ring buffer, and estimating the seasonal component. In particular, the estimation of the seasonal component leverages Seasonal and Trend Decomposition using Loess (STL) decomposition, which breaks down the data into long-term trend, seasonal trend, and residual short-term remainder, a process

elaborated on in Section 2.6

Finally, the power output is adjusted for air density (denoted as $\rho$). A moving average filter is applied to smooth the data and a de-trended plot is generated for better visualisation.

This methodology provides a robust framework for dissecting and evaluating the performance of wind turbines over a specified period and varying time frames considering various performance-altering factors. It allows the identification of seasonal

trends, short-term variations and long-term performance trends, offering valuable insight into wind turbine's performance evolution. Notably, this method is effective without direct wind speed measurements.





## 2.6 Seasonal Trend Decomposition

After calculating the TPI virtual sensor signal from the generator speed-power relationship, Seasonal and Trend Decomposition using Loess is employed to extract valuable insights from the time-series data. STL is a robust and adaptable time-series decomposition technique Cleveland et al. (1990) with widespread applications Sanchez-Vazquez et al. (2012) Hafen et al. (2009) Anderson et al. (2013) Xu et al. (2023) Verbesselt et al. (2010). Decomposition models have otherwise been proposed in the application of wind forecasting Prema and Rao (2015). Its suitability for analysing the wind turbine performance integral stems from the presence of both trend and seasonal components in the data. STL uses LOESS (Locally Estimated Scatterplot Smoothing) to decompose the time series into three main components: long term trend, seasonality, and remainder.

The long term trend component isolates the underlying progression of turbine performance over extended periods. Unwanted seasonal elements and random noise are filtered out, providing a smoothed signal representing factors such as O&M events, cumulative wear and tear of the wind turbine or longer term climatic changes.

The seasonal component captures cyclical atmospheric conditions related effects and variations over multiple years. It effectively attempts to remove the provided signal of the multitude of weather effects such as turbulence patterns and other wind patterns. For this study, the frequency of this component is set to an annual frequency, essentially 'telling' the function where to expect a repeating pattern in the signal, which aligns with the periodicity expected in wind data. This approach reveals performance trends within a calendar year. Notably, STL decomposition does not artificially create seasonality; a missing seasonal component at the fed cycle, or an incorrect frequency specification or absent seasonal pattern, results in a very apparent weak or missing decomposition.

The short term remainder component of the signal encapsulates what is left over from the time series after removal of the long term and seasonal trends and is constituted by erratic or erroneous fluctuations. The variations seen in this component may on occasion be attributed to transient technical issues or daily weather changes or may remain unaccounted-for factors due to immediately unknown causes that may include erroneous data.

The implementation of STL in the methodology was done through MATLAB's "trenddecomp" function The MathWorks, Inc. (2023) applied to the virtual TPI signal generated from the relationship between generator speed and turbine power.

## 2.7 Data Visualisation

Data visualisation plays a crucial role in interpreting the insights derived from the STL decomposition results. A multi-panel figure with a common x-axis (time) and individual y-axes for each panel was employed to effectively present the findings in a comprehensive view. This structure facilitates the simultaneous examination of the trend, seasonal, and remainder components, revealing their interaction over time. This was applied for each wind turbine for the twelve year period of analysis.

The common x-axis, representing time, provides a reference point for all subplots, enhancing the understanding of the concurrent evolution of these components. Each panel of the stacked plot is dedicated to one component of the STL decomposition: long-term trend, seasonal trend, short-term remainder, and the combination of these components (full data, long term + sea-





sonal). By examining patterns within each component, one can determine seasonal and long-term trend impacts on overall
wind turbine performance and identify potential anomalies.

## 2.8 Statistical analysis

Moving beyond visual assessment and accounting for data uncertainty and limited sample sizes for certain categories, the
application of statistical methods is imperative. While observational analysis provides valuable initial insights, it does not fully
account for the intrinsic and nuanced variability within the dataset. The use of statistical tests provides a scientific methodology
to determine the likelihood that observed effects genuinely exist and are not merely coincidental artefacts. In this context the
following describe normality assessment, significance tests, and optimal sample size considerations, to gain a comprehensive
understanding of the dataset and the study's findings.

For the Normality Assessment, the selection of suitable statistical tests for the analysis depends on the distribution charac-
teristics of the 'Difference' values across each event category. The Shapiro-Wilk test Shapiro and Wilk (1965), known for its
efficacy in assessing normality, was employed to test the null hypothesis that these values originate from a normally distributed
population. It is important to note that this test can be sensitive to sample size, especially for very small or very large samples,
with optimal performance for sample sizes around 20-50, its application requires careful consideration. A p-value threshold of
0.05 was used. Results above this threshold do not definitively prove normality but rather as an absence of sufficient evidence
to the contrary, justifying a working assumption of normality.

Following which Significance Tests were carried out to assess whether the performance differences for each event category
were significantly different from zero. Depending on the normality of the data in each category, one-sample t-tests Student
(1908) or Wilcoxon signed-rank tests were utilised. The latter being particularly valuable for non-normally distributed data
Wilcoxon (1945), offering an alternative to parametric tests. This test has been previously employed in studies regarding
condition monitoring and fault detection in wind turbines Dao (2022).

Finally, for the Sample Size Considerations, optimal sample size depends on a desired level of statistical power, the magni-
tude of the effect size to be detected and the variability within the dataset. A power analysis will be conducted Cohen (1988) to
ascertain the sample size necessary to detect a given effect with a specified confidence level. The desired power was set at 0.8,
representing an 80% probability of correctly rejecting the null hypothesis when it indeed is false. The alpha, or significance
level, was set to 0.05 for the alpha level, indicating a 5% risk of erroneously rejecting the null hypothesis when it is true. Par-
265 ticular attention was paid to categories with limited sample sizes (fewer than 20 data points), as these can impact the reliability
of statistical tests. The effect size, indicative of the magnitude of the difference aimed to detect, was estimated from the current
data set using Cohen's d as a metric, alongside measures of variability such as the standard deviation. It should be noted that
assuming sample data as effect size and sample standard deviation as true deviation is more of an exercise in curiosity than a
hard statistical fact. If the effect is larger or smaller than this, the number can changes wildly.



# 3 Results and Discussion

## 3.1 The effects of software updates

Figure 1 and 2 illustrate the evolution of the pitch angle and normalised generator speed, respectively, as functions of power output for the initial decade of a turbine's operation, segmented into annual intervals. The analysis indicates recognisable alterations in the operational trajectory of both parameters within the first half of the observed period. These changes appear to coincide with the rollout of turbine manufacturer software updates while the turbines were under an OEM service contract. Each update likely altered both the pitch versus power and generator speed versus power curves relationships. In subsequent years, without any software updates, the curves stabilised, showing no significant deviations. This supports the hypothesis that these updates were the causal factor in the observed shifts.

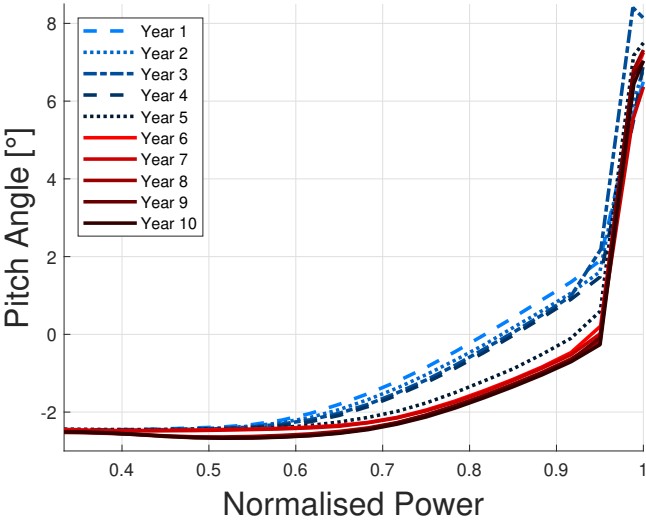

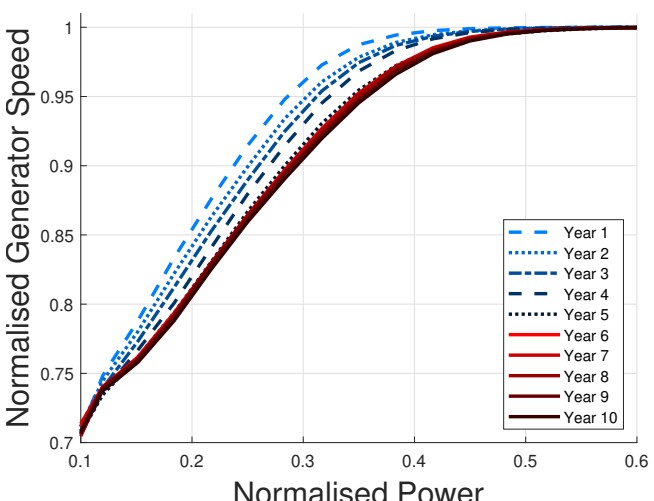

**Figure 1.** Pitch Angle vs. Normalised Power: Yearly Intervals Analysis

**Figure 2.** Normalised Generator Speed vs. Normalised Power: Yearly Intervals Analysis

Beyond the shift itself, these alterations suggest specific impacts. In the higher power range, where the generator speed is constant and the turbine's control solely relies on the pitch, a progressive annual decrease in pitch angle of up to $2^o$ was observed at power levels approaching the upper threshold of the nominal capacity, until year five of operation.

In the lower normalised power range, despite the pitch angle remaining constant and unaltered, the generator exhibited progressively higher relative power outputs at specific normalised generator speeds over the initial five years. This increase is noteworthy, with relative power gains reaching up to 10% of nominal capacity at generator speeds close to the upper limit of the nominal operating range.

As the turbine power increases, the point of maximum thrust, known as the "knee" of the power curve, is reached around the rated power. Here, the turbine blades initiate their pitching, thereby relieving loads on the blades. These findings suggest





that the change in the relation between pitch and generator speed as a function of power is not as a consequence of degradation of the rotor performance, but more likely due to software updates resulting in more aggressive pitch control, increasing blade
loading as they pitched in more frequently. This shift in control strategy boosted power output, albeit potentially at the expense of reduced blade and turbine component longevity. Despite these changes, an Original Equipment Manufacturer (OEM) may describe this adjustment as 'load neutral' or within the turbine's certified design tolerances.

These observations raise questions as to whether such 'uprates' in power are strategic decisions, balancing immediate gains against long-term component durability. One may also consider whether such business-impacting decisions were made solely
by the supplier, or collaboratively with the owner. Consideration must be given to economic aspects, weighing immediate benefits against component lifespan, particularly in light of potentially less favourable future energy tariffs.

Such software updates introduce an additional variable into the complex 'haystack' of factors or uncertainties that can obscure the relatively smaller performance shifts caused by blade erosion. Accurate documenting and logging of these updates, along with their tangible effects on turbine control, is imperative to be factored into assessments. Changes in parameters can
significantly alter conclusions regarding the turbine's response and are critical in the inclusion in holistic long term performance evaluations.

### 3.2   Seasonal Trend Decomposition

The analysis of the time series data for the twelve year period of analysis revealed the distinct trend components influencing the generator speed versus power or TPI characteristics for the twelve wind turbines. Figure 3 exemplifies a single turbine's
decomposition. To reiterate, as detailed earlier in Section 2.5, a TPI decrease reflects improved turbine performance.

The Long Term Trend subplot highlights the fundamental performance trend within the data. While it experienced variations over the analysed time period, the trend remained stable considering its variation between the start and end dates of the time frame. This trend and its correlation with the lifecycle events of the twelve turbines will be further discussed in Section 3.4.

The Seasonal Trend definitively confirms expected cyclic patterns. Notably, seasonality in turbine performance emerges
distinctly with peaks during summer and troughs in winter. A deeper exploration of this trend is undertaken in Section 3.3.

The third subplot of Figure 3 illustrates the Short Term Remaining (STR). This represents the residual fluctuations after accounting for the long-term and seasonal trends, capturing unexpected events or anomalies beyond the other trends.

Lastly, the combined subplot integrates the effects of Long Term Trend, Seasonal Trend, and STR providing a holistic representation of the raw signal over time.

### 3.3   Seasonal influence

Figure 4 depicts the superimposed seasonal trend signals for all twelve turbines. Despite the inherent noise in the data, clear seasonal patterns emerge, characterised by pronounced peaks indicating reduced performance in the summer months and noticeable troughs signifying improved performance during the winter. This counterintuitive direction stemming from the turbine specific approach used in the TPI calculation.





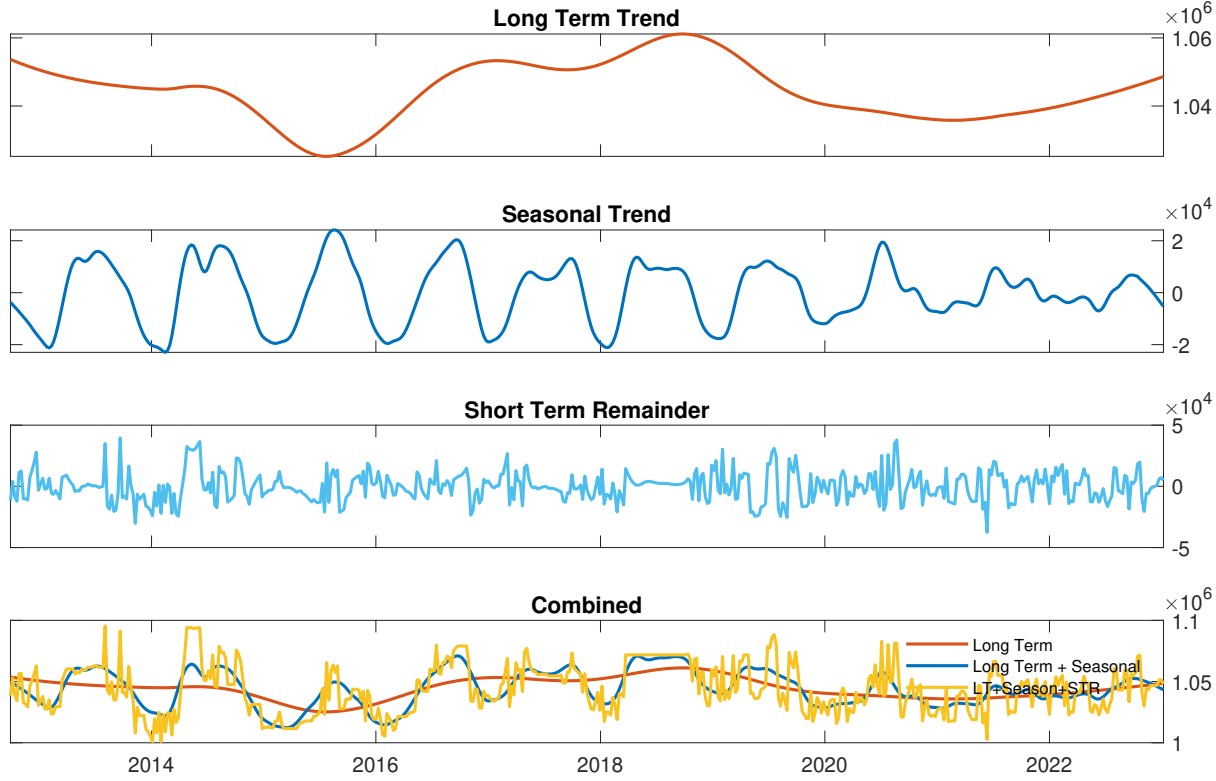

**Figure 3.** Decomposition of a Turbine's Performance Trends Over Twelve Years.

The analysis spanned a dataset spanning several years, focusing on the summer peaks, considering May to September time periods and winter troughs, spanning December to Mid March. Extreme performance values were extracted from these seasons and their distribution was presented using a violin plot Bechtold (2016) Bechtold et al. (2021) (Figure 5) to visualise the distribution of these extreme values across the two seasons. This figure highlights the contrasting oscillatory performance of the turbines across the two seasons.

However, it is noteworthy that the peaks and troughs are inconsistent between the twelve turbines, despite their shared operational environment. This inconsistency may arise from the underlying method of deriving the TPI where a ring buffer, accumulates data before generating the signal. Consequently, such buffering introduces a downstream lag in the TPI signal that depends on, non weather dependent, factors specific to each turbine, such as outages or repairs. The extent of this lag generated by this method is reflected in the varying dates of summer peaks or winter troughs for individual turbines within a given season.

While the ring buffer runs dynamically, individual tuning of buffer parameters for each turbine may offer an opportunity for method refinement (discussed in Section 3.5).

Additionally, Figure 4 suggests muted and diverse summer/winter extremes for all turbines post-2020. This could potentially be caused by greater weather instability in recent years or cumulative ring buffer effects. These factors may also explain the spread of extreme values observed in Figure 5.





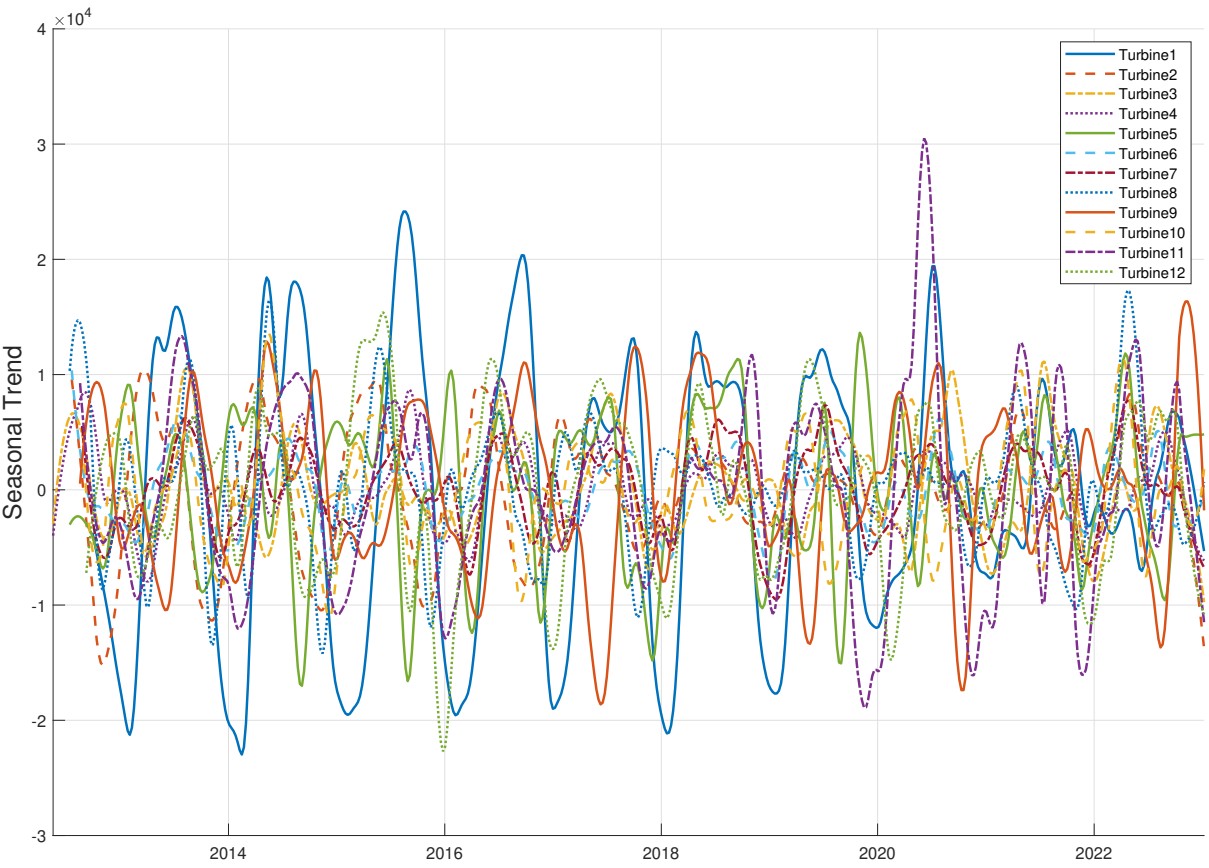

**Figure 4.** Seasonal Performance Patterns: Summarised Seasonal Trends Across Twelve Turbines.

This approach using the TPI demonstrates that distinct seasonal variations in an individual turbine performance can be detected without direct wind speed measurements, or comparing the single turbine to a group or using combined data from multiple sources. Essentially, an individual turbine can distinctly reveal performance fluctuations driven by seasonal dynamics.

However, the evident variance in seasonal performance peaks and troughs across the twelve turbines, also underscore the challenges tied to this approach. Notably, the winter troughs signify a more enhanced or improved performance compared to

the summer peaks. With a sharper understanding of the dates of seasonal variations, one may strategically schedule outages or operations and maintenance, maximising energy production. Furthermore, anticipating periods of high and low generation, strategies can be formulated for energy storage, distribution, and consumption, ensuring grid stability and efficient utilisation.

In the context of climate change, broader performance trends observed across multiple turbines, unrelated to specific events, may be indicative of changing weather patterns affecting turbine performance. This common environmental impact may be

studied to further understand how the performance of turbines is affected by weather. Time series decomposition tools can help differentiate event-driven changes from seasonal, potentially climate-related trends. Future research could include data normalisation to enhance this distinction. Given the shared climatic conditions, normalising the performance of a turbine

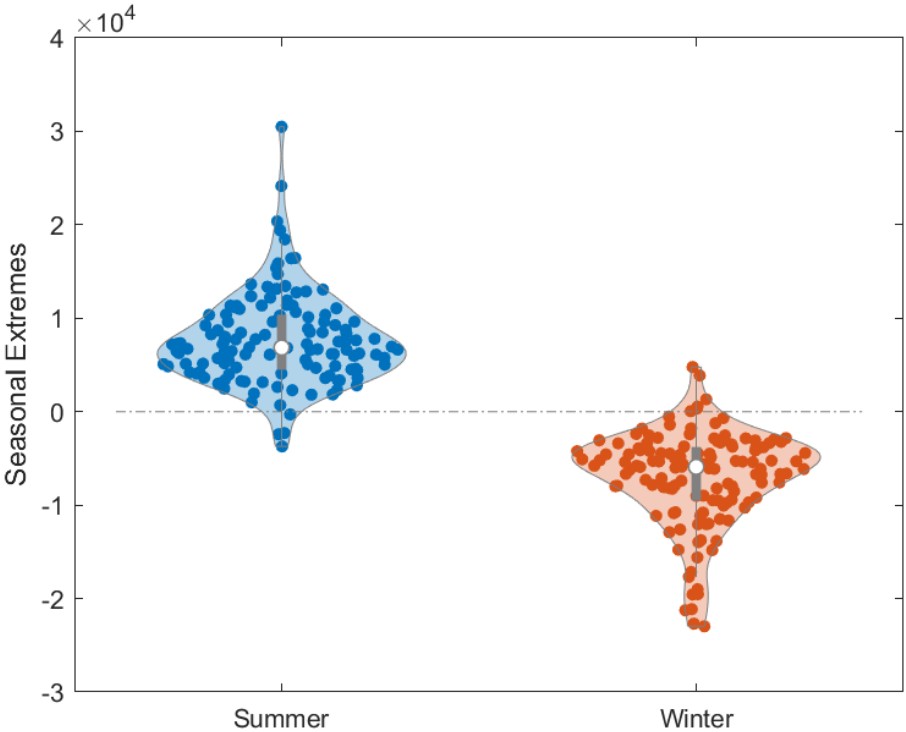

**Figure 5.** Seasonal Performance Extremes for Twelve Turbines: A Comparative Analysis of Summer and Winter Variability. Lower values indicate enhanced performance, while higher values denote reduced efficiency.

against the daily average performance of all turbines could isolate the effects of events from those caused by broader weather influences, offering a purer perspective on the direct impacts of weather on turbine performance.

## 3.4 Long Term Trend

Depicted in Figure 6 are the superimposed long term trend signals of all twelve turbines. Three broad yet distinct clusters among the turbine performances emerge. Turbines sharing similar trajectories of decrease, increase, or variation across the long-term trend component were thus grouped, suggesting possible shared drivers.

– *Group One (light blue): Early Performance Improvement* - Turbines 2, 4, 7, and 12 predominantly exhibit a downward trajectory in TPI, indicating an improvement in performances. This improvement is particularly apparent up to approximately 2017. The early improvement may potentially be associated with the iterative software updates rolled out to these turbines, as discussed in Section 3.1. Post 2017 these turbines performance seems to stabilise and become more consistent, likely due to the end of service contract with the OEM and cessation of further software updates.

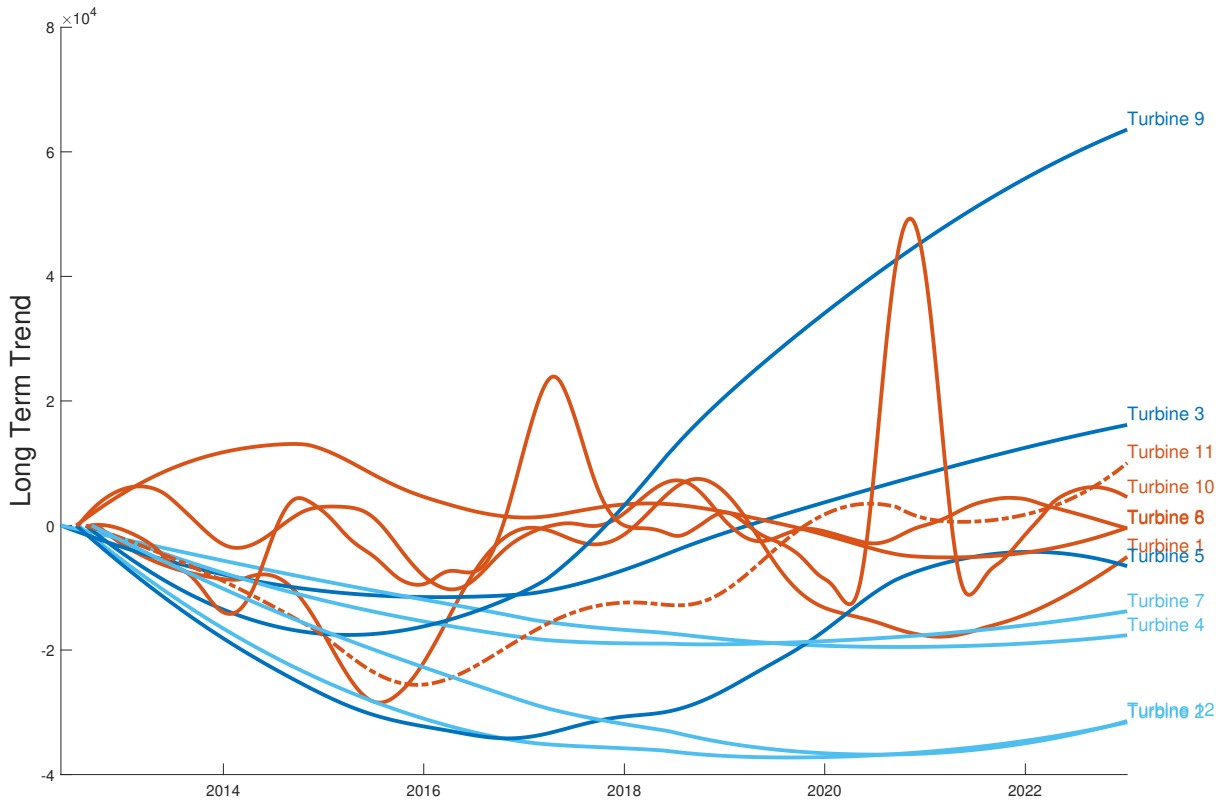

**Figure 6.** Grouped Long-Term Trends in Turbine Performance: Analysis of Shared Trajectories Among Twelve Turbines.

- *Group Two (dark blue): Late Performance Decline* - Turbines 3, 5 and 9 show a similar improvement in performance during earlier years to Group One. However, they subsequently experience a drop with a progression of time. An exception being Turbine 5, that exhibits a later stage improvement and also displays slight performance undulations through its operational lifespan, in a pattern similar to those seen for Group 3.

- *Group Three (red): Synchronised Performance Variability* - Turbines 1, 6, 8, 10, and 11, exhibit predominantly synchronised performance fluctuations, albeit with a lag between them, that may be ascribed to the ring buffer affects. This affect was seen earlier in the seasonal trend for the turbines in Figure 4, in Section 3.3. In the cluster, Turbine 11 notably deviates from the common longer term trend path, exhibiting characteristics of Group One, especially towards the later progression of time. Despite this, most turbines in this group show an improvement in trend in the earlier years relative to Groups One and Two, that may be attributable to software updates. It should be noted, however, that updates for certain turbines may have been applied at differing intervals. In later sections, this shared performance pattern is explored more thoroughly, aiming to correlate it with specific events in the timelines of these turbines. If common events are not identified, it may be plausible that these commonalities are attributed to long term climatic changes, though this would need to be reconciled with Group One's lack of similar effects.





As illustrated in the long-term trend graph of Figure 6, it is evident that the aggregation method and data handling techniques can greatly influence the interpretation of performance trends. The application of a ring buffer, consistent across all turbines
in the analysis, in data processing, while not explicitly shown in the graph, underpins the observed performance trends by controlling the flow and smoothing of the input data. A smaller buffer size provides a detailed, immediate view, ideal for detecting short-term variations and immediate operational issues. On the other hand, a larger buffer size offers a comprehensive overview, highlighting long-term trends and gradual changes but potentially missing brief anomalies. Future studies could involve analysing the effects of different buffer sizes to assess their impact on the long-term performance trajectories, providing
insights into the optimal data processing parameters for trend analysis. This would refine the analytical model, potentially offering a more tailored approach to understanding each turbine's unique performance characteristics.

## 3.5    Influence of erosion, blade enhancements and Operations and Maintenance events

This section aims to uncover correlations between specific maintenance activities, blade modifications, and the observed long-term TPI trends of individual turbines. Efforts taken to compile a comprehensive record of maintenance activities are outlined
Section 2.1. Compiled field maintenance reports and financial accounting data sourced from SAP were amalgamated and overlaid visually on the long-term trend component (Figure 7) for Turbine 1.

To reduce subjectivity in this assessment and observational bias of visual correlations an automated approach was used to quantify the influence of various event categories on the TPI long-term trend gradient. The gradient was calculated between two points: the first two weeks prior to an event, accounting for repair time - and another three weeks after the event to capture
potential trend deviations. Varying the temporal buffers around the events resulted in analogous outcomes for the analysis. It is important to note that the analysis presented here is based solely on the field maintenance reports. The accounting SAP data, despite being a potential source of event information, was not utilised in this analysis due to the incorrect event dates associated with the works. This limitation caused by the temporal discrepancies in the SAP data, is significant as it results in a partial view of the event data, affecting the comprehensiveness and accuracy of the analysis. Figure 8 presents the results as violin plots,
visually summarising the performance impact distribution for each event category.

The plot provides an overview of the performance impact distribution for each event category, deliberately focused on densest region, while retaining the rare (three) outliers in the underlying data. The direction towards the positive signifies a reduction in turbine performance. The white dot represents the median of the distribution, the bar in the centre the interquartile range and the whiskers the remaining distribution. Each shape provides the distribution of individual performance impact events for the
category. The bulge of the shape at different gradients represents the density of event data points at that value. Essentially, the spread of data within a category is easily identified by its shape. Additionally, categories of events that have the most variable effects on turbine performance, and those that tend to have consistently positive or negative impacts can be identified.

Key observations and limitations from this analysis include:

- *Controller PLC parameter or Software Updates (Latest):* Numerous software updates were applied to all twelve turbines
over the first five years of operation. The plot represents the impact of only the last software update applied in the last

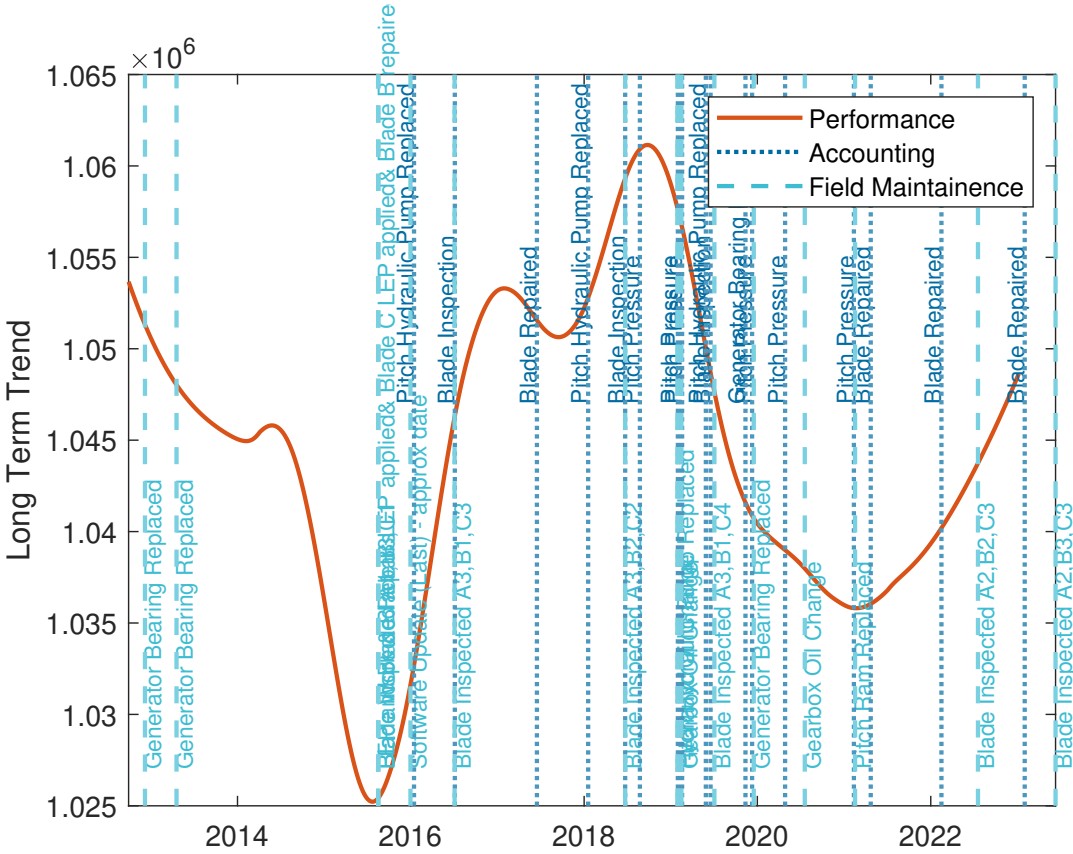

**Figure 7.** Overlay of Maintenance Activities and Blade Modifications on Long-Term TPI Trends.

quarter of 2015. The category shows a relatively narrow distribution that is negatively skewed indicating that the updates often improved turbine performance. The effects were first observed by the change in turbine operational trajectory in Figure 1 and 2. These performance changes align with observations of the Long Term Trend of the turbines in Section 3.4 and Figure 6, although, in the earlier figure, with a wider overview, a somewhat continuous improvement over the

first five years of the turbine life, was observed for the numerous iterative updates.

– *Blade Repairs (all):* Interpreting the impact of blade repairs, it is important to note, that new blades may not always precisely match the aerodynamicist's intended profile due to manufacturing variances - tolerances for which are treated as intellectual property by manufacturers - which subtly affecting baseline performance Ernst et al. (2014) Loeven and Bijl (2008). Regarding uptower repairs, a wider distribution with a slight positive performance bias suggests that such

repairs can sometimes lead to a decrease in turbine performance. However, the variance is wide, indicating that the impact can be inconsistent. This is a curious and unexpected result. However, digging deeper one must acknowledge that not all blade repairs have a positive influence on the blade aerodynamics. Blade repairs often aim to stop or prevent progression of structural damage to a blade leading edge. The blade leading edge is sanded and filler applied after which leading edge



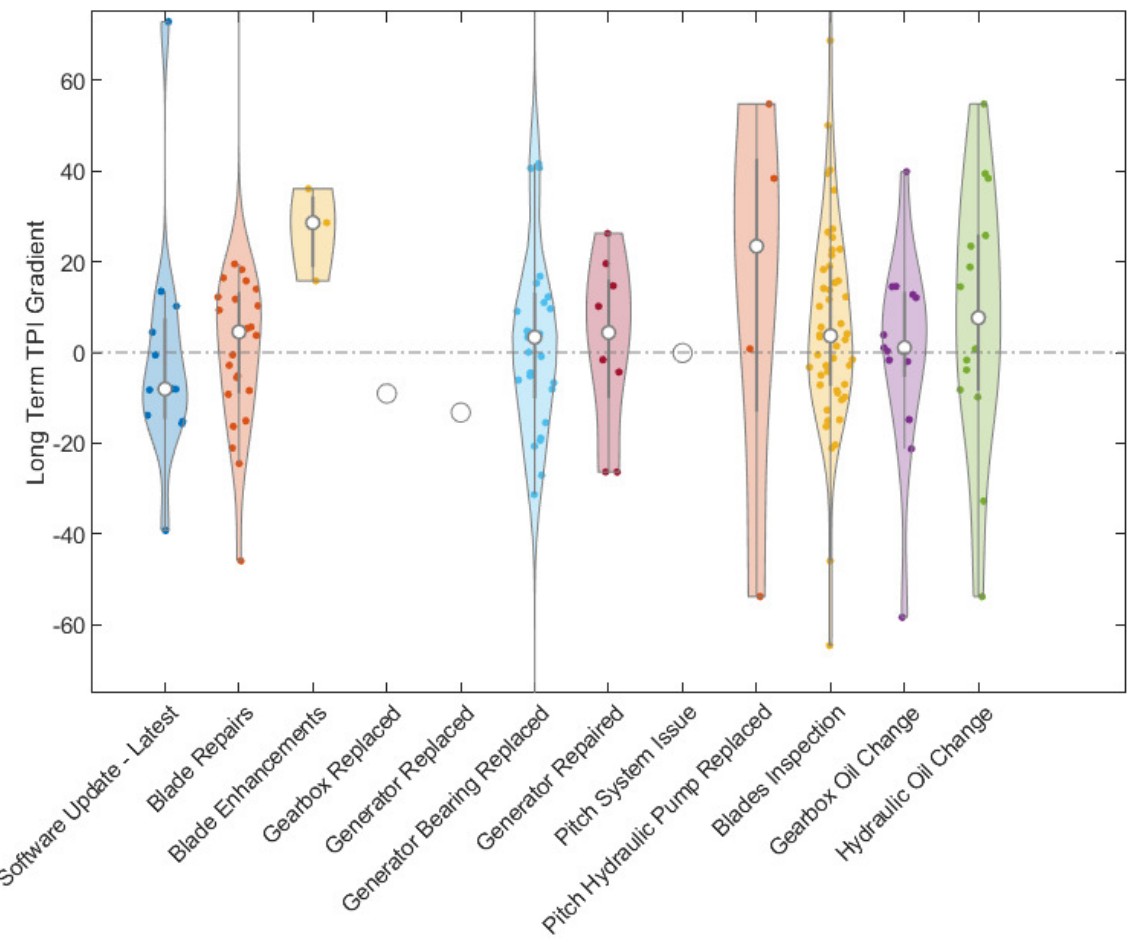

**Figure 8.** Event Impact on TPI Trend: Violin plots summarising the performance impacts post-events, derived from gradients calculated around event timings based on field reports, excluding SAP data due to date discrepancies.





protection (LEP) may be applied. The task is labour-intensive requiring trained, skilled personnel, often working under sub-optimal conditions, challenges that are amplified in offshore uptower environments. These repairs often result in the blade leading edge no longer matching its original and intended aerodynamic profile, resulting in a reduction in efficiency Katnam et al. (2015). In the case of the underlying dataset represented here, all blade repairs including those that are known to be defective and thus detrimental to turbine performance are included. It is therefore prudent to break down the category of blade repairs into those that result in a cleaner blade profile and those that are likely have degraded the turbine performance. Additionally, it is crucial to consider the seasonal timing of blade repairs, predominantly conducted during the summer months, which may inadvertently introduce a bias into the analysis. This seasonal scheduling of maintenance activities, driven by operational and logistical considerations, especially in an offshore environment, could skew the observed performance trends, creating an uncorrected seasonal bias that influences the interpretation of the TPI's long-term performance trajectory.

– *Blade Inspections:* Although inspections may be a precursor to repairs, this category comprises of events or data points when the turbine blades were simply inspected and no works were conducted. The category can serve as somewhat of a control, since no actual physical changes were made to the turbine blade. The very wide distribution with a slight negative skew suggests that blade inspections can sometimes lead to improved performance, but not consistently. The degree of slight negative skew may be interpreted as perhaps the uncertainty of the method. This is since there should in fact be no expected change in turbine performance and is, of course, not reflecting reality. However, it is worth speculating that during some of these inspections, the work instructions may have encompassed a pitch re-calibration, potentially influencing these outcomes. While this remains speculative, such re-calibrations, if they occur, could inadvertently impact performance, thereby slightly altering the expected 'no change' scenario post-inspection. Similar to blade repairs, the timing of these inspections, often aligned with the summer months for operational convenience, introduces an uncorrected seasonal bias into the analysis.

– *Blade Enhancements:* A number of turbines in the wind farm were furnished with third party Gurney flaps and vortex generators as an experiment to understand their impact on the turbine performance. Three of these turbines were in population under investigation. The efficacy of these enhancements, hinges on their positional design tolerance and consequently precise application, which is notably more challenging in uptower installations than in controlled factory conditions, potentially impacting the observed performance outcomes. These blade enhancements were not applied in collaboration of the OEM and thus there was no associated controller changes made - software updates being under the exclusive mandate of the OEM and integral to the turbine's certification. It was noted that the wind anemometer of these turbines showed a step change in measurement of wind speed compared to their neighbours in the same row, which would have a direct impact on the turbine's operational setpoint, considering it's control utilises the anemometer for its operation. The upshot of these changes to the aerodynamic profile of the blade are that the distribution is positively skewed with a relatively narrow distribution, indicating that blade enhancements often lead to a decrease in performance for the three turbines. Although, one should consider that this is on the basis of a very small population size.





– *Gearbox, Generator replacement:* These categories both show a slight improvement in turbine performance as would be expected when a faulty major component such as these is replaced. Although, since these categories each have a population of one. The high uncertainty should be considered while making conclusions.

– *Generator Bearing Replaced, Generator Repaired, Pitch Hydraulic Pump Replaced:* These categories show, to varying degrees a reduction in turbine performance. The exclusion of some additional generator bearing faults, as noted in the accounting SAP data but not time-matched with or entirely absent in the repair reports, highlights a limitation of the analysis.

– *Pitch System Issue*: This category shows a neutral impact. Again, the high uncertainty due to it being a population of one should be considered while making conclusions. However, again, an important consideration in the interpretation of these results is the discrepancy in the recording of events between different data sources. Notably, numerous pitch ram replacements were identified in the accounting SAP data, which were not captured in the field maintenance reports. This discrepancy is significant, as the replacement of a critical component such as the pitch ram could have a substantial impact on the turbine's performance, potentially affecting its ability to correctly adjust blade pitch in response to wind conditions. The absence of these events in the repair reports introduces a limitation in the analysis, as the influence of pitch ram replacements on turbine performance remains unaccounted for.

Additionally, one may consider the mechanical integration of the blade root end, pitch bearing, and hub. Over time or the blade root flange surface could deviate from design specifications such as its flatness and perpendicularity. Such deviations could potentially compromise the turbine's ability to accurately adjust the blade pitch, further influencing performance outcomes.

– *Gearbox Oil Change, Hydraulic Oil Change:* These are both maintenance categories that would not be expected to have an impact on the turbines performance. This expected result is seen in case of the former, Gearbox oil change, where the median value is very close to zero, however, with a wide variance but normal spread in results.

Hydraulic Oil Change shows a reduction in performance but again with a wide variance but normal spread. Hypothesising, this is work where the thoroughness of the employed maintenance procedures, might influence outcomes. Particularly regarding the degassing of hydraulic fluid, failing which can lead to a significant loss of performance. However, this connection is speculative and also highlights the need for further investigation and refinement of the method employed by the study.

The dataset for Blade repair contained all blade repairs, including those that were known to have a detrimental impact on the turbine performance. Here, in Figure 9, this repair is split into two populations, where one is known to be defective LEP and the other to be the remaining LEP application and repairs. It should be noted that the latter population is still likely to contain faulty applications of LEP and that this analysis does not account for the uncorrected seasonal bias, discussed earlier regarding blade repairs.

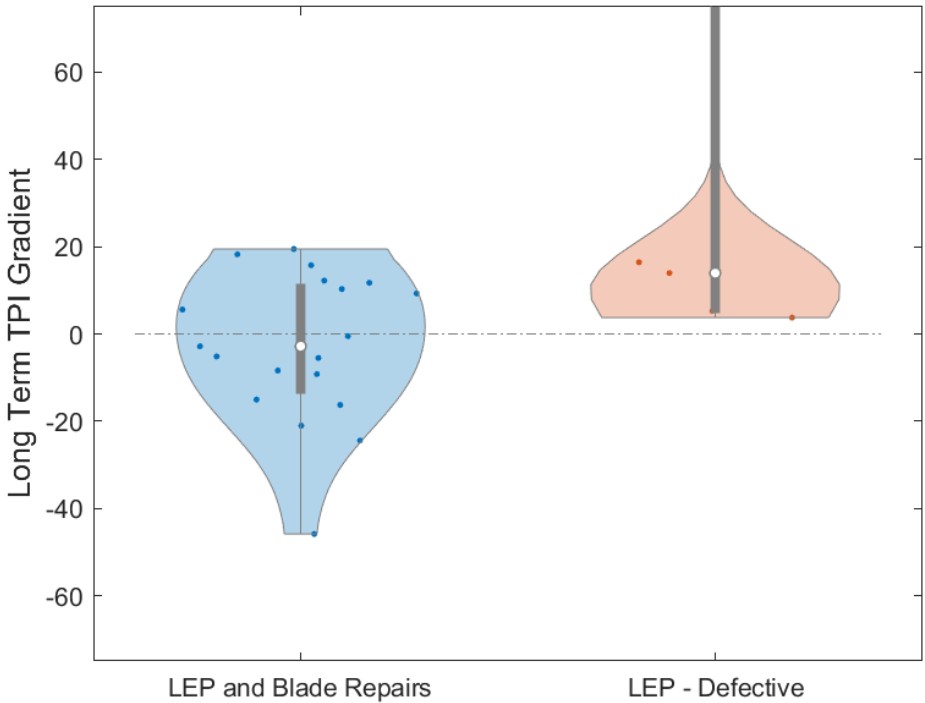

**Figure 9.** Dichotomy in Blade Repair Outcomes: Evaluating Performance Impact of Defective LEP vs. Repairs and Non-Defective LEP Applications. Note: Includes potential unaccounted faulty LEP in the latter category without correcting for seasonal biases.

– *LEP - Defective:* From the dataset of Blade Repairs (all), those of a known LEP repair to have had a detrimental effect on the blade leading edge were extracted. Blade repair technicians reported that this LEP left the blade in a tacky and sticky condition gathering contaminants within the first few days and weeks of application, with the strong recommendation to cease all further application of the product. The results of the impact on performance of the LEP show a strongly positive skew indicating a detrimental impact on the turbine performance. Note that a single outlier at 279.5, while retained in
the data, has been omitted from the plots focus.

        – *LEP and Blade Repairs (Remaining):* This is the remainder of the the complete Blade Repair dataset after removal of the known detrimental or defective LEP. A slight negative bias suggests that blade repairs can sometimes lead to an improvement in turbine performance. However, the variance is wide, indicating that the impact can be inconsistent. This inconsistency may be attributed to the various practical challenges associated with blade repairs and LEP application,
discussed earlier. These earlier described factors can influence the aerodynamic efficiency of the blade post-repair, once it deviates from its intended profile, potentially leading to performance that may be worse than certain levels of erosion.





From Figure 8 one can observe which event categories tend to have a stronger or weaker impact on turbine performance. A concentration of event data points in a further away from zero, such as in the case of the positive gradient of blade enhancements indicates a greater reduction in turbine performance.

Although corrections were made for atmospheric conditions, including air density, temperature, and humidity, the analysis does not consider the variability of other atmospheric conditions and factors such as turbulence, wind shear or veer. These factors, despite being excluded, are crucial for a comprehensive understanding as they can significantly influence the power output of wind turbines Wharton and Lundquist (2012) Murphy et al. (2019) St Martin et al. (2016) Kim et al. (2021). Variations in these conditions can lead to discrepancies between actual and expected performance, potentially confounding the analysis

if not correctly factored in. For instance, the wind profile's alignment with the rotor plane directly influences the efficiency of energy capture Wan et al. (2015). Yaw misalignment, can also lead to performance degradation. A yaw offset can cause uneven loading on the turbine blades, increased mechanical stress, and reduced energy capture efficiency van Dijk et al. (2016). This misalignment may be caused maintenance issues or sensor inaccuracies, and can result in a sustained loss of performance. Moreover, the use of nacelle direction as a surrogate for wind direction, influenced by the turbine's control algorithm hysteresis,

further complicates the interpretation of findings.

The integrity of a turbine's foundational and structural elements, such as its blades and tower, are paramount to its performance. Over time, shifts or deterioration in these structures can subtly influence performance metrics, potentially skewing data analysis if not accounted for. Further, it is important to acknowledge the role of dynamic forces and environmental conditions on turbine blade performance. Centrifugal force, wind loading, and the thermal expansion of blades, depending on environ-

mental conditions, can also influence the Angle of Attack (AOA), twist and prebend of blade profiles, consequently altering the blade polars Loeven and Bijl (2008). Additionally, the health of mechanical components like gearboxes and drivetrain is crucial; wear and tear on these components can lead to performance degradation that maintenance records may not immediately reflect. Sensor accuracy and calibration are also critical factors. The drift in precision and calibration of instruments like anemometers and power meters can significantly impact the reliability of the data collected Pindado et al. (2012). Considering

these aspects, it is evident that there may be a myriad of other influential events and factors that have not been captured in the analysis, which could affect the results. The omission of such details underscores the complexity of turbine performance analysis and the necessity for comprehensive data integration and meticulous scrutiny to inform accurate interpretations.

The analysis of turbine performance was further complicated by the imprecise temporal recording of O&M events and the notable absence of crucial event records. The investigation was hindered by the discrepancies between the financial accounting

SAP database and the repair reports; the SAP database logged several critical events regarding major components, such as pitch ram replacements, yaw faults, generator bearing faults (CMS Comms), and a wind anemometer replacement which were absent in the repair documentation. This discrepancy likely arose because the data foundation is SAP accounting reports, where dates often correspond to billing rather than the actual repair dates. Consequently, there are likely other significant O&M events affecting the twelve turbines that were crucial to the analysis but were not recorded. This gap between event occurrence

and its documentation lead to inconclusive results, as the analysis may have have overlooked key incidents that substantially





impacted turbine performance, skewing the understanding of cause-and-effect relationships within the operational data and having tangible implications on the accuracy of interpretations of trends seen in Figure 8.

However, the inherent date lag, together with the fixed buffer size designated for data smoothing, introduces an element of uncertainty in this alignment. The buffer size, set to process 1000 data points at a time, influences the temporal resolution of

the analysis. Especially, given the variability in the lag over time and between turbines, which can span up to several weeks or even months, identifying specific O&M events becomes challenging. For instance, if a turbine exhibits a sudden change in its performance, attributing this decline to a specific event becomes ambiguous when there's a potential delay of of an unknown period in the signal. The buffer's role in data processing could inadvertently synchronise disparate events or mask the true temporal sequence, thereby complicating the interpretation of causality within the operational timeline of the turbines. Such

delays can lead to misinterpretations, where an O&M activity may be mistakenly attributed with a performance change that actually occurred due to a completely different event, or combination of sequentially occurring events. This is particularly evident in the noise seen in Figure 7, where the proximity of numerous events potentially distorts the clarity of cause and effect, diminishing the reliability of conclusions drawn from the data.

A salient consequence of the date lag, as elaborated in the Section 3.3, is its potential impact on the accuracy and reliability

of the long performance trend signal. When assessing the TPI over an extended period, it was attempted to align these observed trends with specific operational and maintenance (O&M) events to derive insights into changes turbine performance.

In light of the errors introduced by the variable lag, it is imperative to account for its influence when interpreting the long-term trend data for an individual turbine and which becomes more challenging when comparing data across different turbines. Potential solutions might involve refining the data collection methodology, employing advanced analytical techniques to correct

for the lag, or using supplementary data sources to corroborate the observed trends.

Moreover, for wind farm owners and operators, the lack of temporal precision of conclusions has tangible implications. Strategic decisions, such as scheduling maintenance, outages, resource allocation, or forecasting of energy outputs, rely heavily on the accuracy of such long-term trend data. Inaccuracies or uncertainties can lead to inefficiencies, increased operational costs and missed opportunities for performance optimisation.

Overcoming these challenges may allow further work on comparative analysis, where one can also conduct a comparative analysis between turbines. In cases such as when a specific event leads to a performance drop in one turbine but not in another. In such examples it would be compelling to investigate the underlying reasons. The reasons may vary from difference in maintenance history, age, to phenomenon such as difference in software updates.

A further avenue of investigation may be temporal analysis. Analysing the data over time may provide insights into whether

there are specific periods where the impact on turbine performance is more pronounced. This may be achieved by plotting the gradient over time for each category of event.



## 3.6 Statistical Analysis

### 3.6.1 Normality Assessment

The normality tests were conducted using the 'swtest' function available in MATLAB, as implemented by BenSaïda (2024).
The Shapiro-Wilk test outcomes, shown in see Table 1, indicate varied distribution patterns across categories. Notably, categories such as blade enhancements and generator repaired did not show sufficient evidence of non-normality. In contrast, categories like blade repairs and generator bearing replaced suggest non-normal distributions. Categories yielding inconclusive results, marked by NaN p-values, highlight the need for cautious result interpretation due to insufficient data points. These findings guide subsequent analyses: data that does not show sufficient evidence of non-normality is amenable to parametric
tests, while non-normal data requires non-parametric methods.

| Category Description | p-value | Test Result |
|---|---|---|
| Controller PLC parameter or Software Update - Latest | 0.0063 | Fail |
| Blade Repairs | $1.23 \times 10^{-7}$ | Fail |
| Blade Enhancements | 0.7122 | Pass |
| Gearbox Replaced | NaN | Inconclusive |
| Generator Replaced | NaN | Inconclusive |
| Generator Bearing Replaced | $3.00 \times 10^{-8}$ | Fail |
| Generator Repaired | 0.4156 | Pass |
| Pitch System Issue | NaN | Inconclusive |
| Pitch Hydraulic Pump Replaced | 0.5850 | Pass |
| Blades Inspection | $1.32 \times 10^{-9}$ | Fail |
| Gearbox Oil Change | 0.0467 | Fail |
| Hydraulic Oil Change | 0.9054 | Pass |
| LEP and Blade Repairs | 0.2332 | Pass |
| LEP - Defective | 0.0023 | Fail |

**Table 1.** Results of Shapiro-Wilk Normality Test

### 3.6.2 Significance Tests

Significance test results are outlined in Table 2. Notably, the category of Blade Enhancements showed a statistically significant difference from zero, indicating that events in this category have a measurable impact on turbine performance. It is worth considering the potential impact of experiment-wise error – the increased likelihood of false positives when conducting
multiple tests. A conservative approach, such as a Bonferroni correction, could be explored to address this, which would significantly lower the threshold for statistical significance. While blade enhancements do show significant changes, this is only if experiment-wise error is not corrected for. Essentially, without correcting for multiple comparisons, the risk of Type I errors





(false positives) increases, potentially leading to overstating the significance of some findings. This suggests the need for a larger dataset to reinforce and enhance the reliability the findings - especially for categories where sample sizes were limited.
However, the employed conservative approach is appropriate given the study's assumptions and the exploratory nature of this analysis.

| Category Description | p-value | Significance |
| --- | --- | --- |
| Controller PLC parameter or Software Update - Latest | 0.380 | No |
| Blade Repairs | 0.530 | No |
| Blade Enhancements | 0.046 | Yes |
| Generator Bearing Replaced | 0.658 | No |
| Generator Repaired | 0.777 | No |
| Pitch Hydraulic Pump Replaced | 0.537 | No |
| Blades Inspection | 0.062 | No |
| Gearbox Oil Change | 0.787 | No |
| Hydraulic Oil Change | 0.353 | No |
| LEP and Blade Repairs | 0.493 | No |
| LEP - Defective | 0.063 | No |

**Table 2.** Results of Significance Tests for Performance Differences

The category of Blade Enhancements showed a statistically significant difference from zero, indicating that events in this category have a measurable impact on turbine performance. It is worth considering the potential impact of experiment-wise error – the increased likelihood of false positives when conducting multiple tests. A conservative approach, such as a Bonferroni
correction, could be explored to address this, which would significantly lower the threshold for statistical significance. While blade enhancements do show significant changes, this is only if experiment-wise error is not corrected for. Essentially, without correcting for multiple comparisons, the risk of Type I errors (false positives) increases, potentially leading to overstating the significance of some findings. This suggests the need for a larger dataset to reinforce and enhance the reliability the findings - especially for categories where sample sizes were limited. However, the employed conservative approach is appropriate given
the study's assumptions and the exploratory nature of this analysis.

Other categories, such as Software Update - Latest and Blade Repairs, did not exhibit statistically significant differences, suggesting that events in these categories may not have a distinguishable impact on performance. In the case of software updates it must be acknowledged that only the approximate date of the last update was logged, which likely influenced the outcome.

It is important to acknowledge the limitations inherent in these tests. Certain categories like gearbox replaced and generator replaced yielded inconclusive results due to the insufficiency of data, highlighting the necessity for cautious interpretation and potentially more extensive data collection for these specific categories. Additionally, this analysis operates under the assumption that events within each category are mutually exclusive for simplicity. However, this assumption does not accurately





represent the complex interactions or compounded effects of multiple events. Moreover, the impact of external factors, pre-
viously discussed, such as weather conditions or the ageing of turbine components, although not captured in this data, likely
significantly influence turbine performance.

### 3.6.3 Sample Size Considerations

Required sample sizes in Table 3 highlight the disparity across event categories. Notably, blade enhancements has a small
required sample size due to a larger, easily detected effect, whereas gearbox oil change requires a substantially larger sample
size, pointing to either negligible effects or intrinsic high variability within this category, necessitating a large sample for
detecting a significant effect. For categories with an 'NaN' result, such as generator replaced, it suggests that the sample size
calculation was not feasible. Indicating impracticalities in sample size calculation due to no sample standard deviation.

| Category | Req. Sample Size | Req. Sample Size (Log Scale) |
| --- | --- | --- |
| Controller PLC parameter - Latest | 13456 | $10^4$ |
| Blade Enhancements | 2 | $10^0$ |
| Gearbox Replaced | 0 | 0 |
| Generator Replaced | NaN | - |
| Generator Repaired | 820 | $10^3$ |
| Pitch System Issue | NaN | - |
| Pitch Hydraulic Pump Replaced | 87 | $10^2$ |
| Gearbox Oil Change | 1206830 | $10^6$ |
| Hydraulic Oil Change | 119 | $10^2$ |
| LEP and Blade Repairs | 306 | $10^3$ |
| LEP - Defective | 29 | $10^1$ |

**Table 3.** Required Sample Size for Statistical Power with Logarithmic Scale Comparison

These sample size estimates serve as valuable guidelines, the sample size estimates should be anticipated as conservative
orders of magnitude, indicating the quantity of data required to confidently detect significant effects in each category. As-
sumptions have been made such as normality for all distributions in the power study whilst there is evidence to the contrary.
Collecting data of these sizes will enhance the power of the statistical tests and bolster the reliability of the results. How-
ever, practical considerations, such as logistical feasibility and resource availability, must also be factored into any decisions
regarding further data collection efforts.

## 4 Conclusions

This study aimed to quantify energy losses in multi-megawatt wind turbines using SCADA data, focusing on the effects of
leading edge roughness. However, significant uncertainties in the data posed major challenges to accurately quantifying aero-





dynamic losses, hindering the study's original goal. Although, it successfully highlighted the complexity involved in such analyses and also brought to light the significant challenges in accurately quantifying the aerodynamic efficiency reductions and subsequent energy production losses caused by LER. A paramount challenge revealed is the accurate quantification of these losses, hindered by the dynamic nature of wind and compounded by operational constraints such as limited data availability, especially in documenting operations & maintenance events, and the quality of the available data. In response, a novel methodological framework was developed, merging SCADA data with detailed O&M records to assess the impact of blade aerodynamic modifications. This innovative approach aims to isolate the effects of these modifications amidst a multitude of factors influencing turbine performance.

The study confronted significant hurdles in accessing comprehensive, high-quality data, a persistent challenge in wind turbine performance analysis. OEM's often restrict data access to safeguard intellectual property, thus limiting in-depth analytical possibilities. Moreover, the opaque nature of turbine control systems further complicates the understanding of turbine behaviour under diverse conditions. Additionally, the investigation uncovered distinct seasonal trends with performance reductions during warmer months and improvements in colder months, demonstrated through overlaid seasonal trend signals. This trend, confirmed by the overlaid seasonal trend signals of all twelve turbines, validates the viability of the suggested methodological framework. The clear emergence of these seasonal patterns, despite the inherent noise in the data, underscores the crucial influence of atmospheric conditions on turbine efficiency and emphasises the need for adaptive maintenance strategies

Software updates, particularly in the initial operational phase of the turbine, were found to be the very likely reason to significantly influence the performance, enhancing power output but also raising concerns about their long-term impact on turbine component's operational lifespan. The study also determined that blade repairs and enhancements have variable impacts on turbine performance. Completing repairs closer to the original aerodynamic design typically improved performance, whereas deviations, such as those by faulty LEP repairs or introduction of third party blade enhancements, without appropriate controller adjustments led to reduced efficiency. Thus underlining the importance of maintaining aerodynamic integrity in blade maintenance procedures.

Rigorous statistical analysis helped distinguish impacts attributable to specific maintenance activities, such as blade enhancements, on turbine performance. These finding emphasises the necessity of a substantial volume of events to establish confidence in results, highlighting the importance of statistical validation in performance analysis. Correlating O&M events with performance data trends revealed discernible effects on turbine performance, yet challenges in establishing precise cause-effect relationships arose due to temporal lags, seasonal biases of occurrence certain events categories and data recording inaccuracies, calling for careful data interpretation. The study thus stresses the critical need for maintaining accurate and comprehensive records.

The diverse long-term performance trends identified among different turbine groups suggest responses to operational, maintenance, and environmental factors or, alternatively, may be attributed to data collection inaccuracies This necessities a customised analytical approach for each turbine or group, tailored to their specific operational context. Future research should emphasise enhanced tracking of O&M events, refinement of data analysis methods, addressing data gaps, and advancing analytical techniques. This may include application of machine learning algorithms for more nuanced insights as well a longitudinal



studies which are crucial for understanding the effects of ageing and environmental changes on turbine performance. Broader data access negotiations with OEMs, exploration of turbine control systems, integration of comprehensive environmental data, and an economic analysis of maintenance strategies are also vital for advancing wind turbine performance optimisation.

*Author contributions.* Tahir Malik was the primary researcher, responsible for the conception of the study, all experimental work, data collection and analysis and the drafting of the manuscript. Christian Bak, as the PhD supervisor, provided oversight, theoretical support and guidance in refining the research methodology and manuscript.

*Competing interests.* The author Tahir H. Malik has received his PhD funding and is by employed by Vattenfall

*Acknowledgements.* We extend our gratitude to our countless colleagues at Vattenfall for their invaluable expertise in wind turbine engineer-
660 ing, which significantly enhanced the analytical depth of our study. Additionally we acknowledge the support of Vattenfall, particularly for financing this study, as well as access to essential wind turbine datasets and resources, enabling a more comprehensive analysis.

We also acknowledge the assistance of the AI language model by OpenAI OpenAI (2023) for its contribution in refining the manuscript.

Our thanks go to Reese Wayne Davies, David C. Manicaci and the team Sandia National Laboratories for their support and constructive feedback.





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
