# Peer review of "Full-Scale Wind Turbine Performance Assessment using TPI: A Study of Aerodynamic Degradation and Operational Influences"

_Wind Energy Science, 2024_

## Referee Comment (RC1)

The manuscript addresses a relevant topic in the wind industry, providing a novel methodology to assess the impact of blade aerodynamic modifications on turbine performance, using SCADA and O&M data. The paper highlights the complexity of the study and the challenges encountered in completing the initial goal of the study. The paper is well structured and in general clear - even though some details can be improved to facilitate the readability of the paper (see my comments attached). Otherwise, very good job!

**General comments:**
- When using an in-text citation where the author's name is not part of the sentence, put the name of the author and year in parentheses to facilitate readability. See the in-text citation Copernicus guidelines.
- I understand that you're using a MATLAB framework to compute the STL long term trend component, short term trend component and seasonal trend. Would it be possible to elaborate on the methodology (driving equations, necessary data inputs or other)?

**Specific comments**:
**Line 4:** Add what PLC stands for.
**Line 16-18:** "Predictions suggest […] various publications" This sentence lacks a bit of clarity, please reformulate.
**Line 34:** Add what CMS stands for.
**Line 67:** I understand that the WTG type and site are confidential. Would it be possible however to indicate the average wind speed at the site and the nominal power of the wind turbine? That would make your results better comparable/ reusable in future related work.
**Line 117-119:** Do you know if a resolution of 10-minutes which is usually provided and stored by SCADA systems would have been enough for the time series of wind speed, nacelle direction, temperature etc.?
**Line 123-124:** Do you mean the dynamic yaw misalignment by "the turbine control algorithm hysteresis"? It's maybe worth mentioning the "static yaw misalignment" on these turbines, that would result in a constant error in the wind direction.
**Line 213:** Consider defining LOESS at its first occurrence (line 198).
**Line 262**: Is 0.8 a probability or power?
**Figure 3:** Add a label for the y-axis.

**Other remarks:**
- Have you considered applying the model on different study cases? It would be interesting to use data from other wind farm(s) to see if the how the model behaves with different assumptions.

---

## Author Comment (AC2)

**General comments:**

- When using an in-text citation where the author's name is not part of the sentence, put the name of the author and year in parentheses to facilitate readability. See the in-text citation Copernicus guidelines.

➤ Corrected throughout paper.

- I understand that you're using a MATLAB framework to compute the STL long term trend component, short term trend component and seasonal trend. Would it be possible to elaborate on the methodology (driving equations, necessary data inputs or other)?

➤ Thank you for your comment. We acknowledge the importance of elaborating on the STL decomposition methodology. While the fundamental equation is referenced as per Cleveland et al. (1990), we have now included a concise explanation of the STL components (trend, seasonal, residual) and their significance within the context of our analysis in the manuscript:

"Following the methodology described by Cleveland et al. (1990), STL decomposes a time series into three components: seasonal, trend and residual. This decomposition is mathematically represented as follows:

$$Y_t = T_t + St + R_t$$

where $Y_t$ denotes the observed data at time *t*, $T_t$ is the trend component, $S_t$ is the seasonal component and $R_t$ is the residual component."

The elaboration has now been added to the paper.

Given the established nature of the STL method and the comprehensive details provided in Cleveland et al. (1990), we have opted to avoid a theoretical discussion to maintain the paper's focus and brevity. However, to address your query, we have explicitly stated in the methodology section that our STL analysis utilises weekly buffered time series data of the area under the curve for generator speed versus power. This ensures consistent and robust input quality for our analysis.

**Specific comments:**

Line 4: Add what PLC stands for.

➤ Corrected

Line 16-18: "Predictions suggest [...] various publications" This sentence lacks a bit of clarity, please reformulate.

➤ Reformulated

"Erosion has the potential to cause significant AEP losses, with some studies predicting reductions of up to 7%"

Line 34: Add what CMS stands for.

➤ Added

Line 67: I understand that the WTG type and site are confidential. Would it be possible however to indicate the average wind speed at the site and the nominal power of the wind turbine? That would make your results better comparable/ reusable in future related work.

➢ Information added

"However, to provide context for the results, the average wind speed at the site is approximately 8.6 m/s, the wind turbine has a nominal power between than 2 and 3 MW and is a horizontal-axis, three-bladed model."

Line 117-119: Do you know if a resolution of 10-minutes which is usually provided and stored by SCADA systems would have been enough for the time series of wind speed, nacelle direction, temperature etc.?

➢ This is a very good suggestion. We have not tested the TPI method with the standard 10-minute intervals commonly used by SCADA systems. However, we have submitted a paper: https://doi.org/10.5194/wes-2024-35 , that explores the impact of varied time averaging in the simulation environment. Future measurement studies could indeed explore this to compare its effectiveness against the higher resolution data used in our current findings.

Line 123-124: Do you mean the dynamic yaw misalignment by "the turbine control algorithm hysteresis"? It's maybe worth mentioning the "static yaw misalignment" on these turbines, that would result in a constant error in the wind direction.

➢ Improved text:

"Moreover, the use of nacelle direction as a surrogate for wind direction introduces additional complexity. This complexity arises from both the potential for dynamic yaw misalignment, influenced by the turbine's control algorithm hysteresis and the possibility of static yaw misalignment, which could result in a constant offset in wind direction measurements."

Line 213: Consider defining LOESS at its first occurrence (line 198).

➢ Amended

Line 262: Is 0.8 a probability or power?

➢ In the context of our study, "0.8" refers to the statistical power of the test, which is indeed a type of probability. It represents an 80% probability of correctly rejecting the null hypothesis when it is false, thus ensuring robust detection of true effects within the data.

Furthermore, the method "Statistical Analysis" section has been improved for clarity. Please see extended response to Anonymous Referee #2's question.

Figure 3: Add a label for the y-axis.

➢ Regarding the y-axis labelling in Figure 3: In our design choice, we opted to position the labels for this particular plot as titles for each subplot to direct reader focus on the key trends depicted. In the belief that this aids in better understanding the data, but we are open to reconsidering this format based on your feedback to improve figure readability.

**Other remarks:**

- Have you considered applying the model on different study cases? It would be interesting to use data from other wind farm(s) to see if the how the model behaves with different assumptions.

➢ Thank you for suggesting the extension of the TPI method to additional wind farms. Indeed, we have already tested the TPI on turbines from another OEM and observed promising results. For further details on this, please refer to our article currently in pre-print and awaiting peer review, available here:

https://doi.org/10.5194/wes-2024-49

---

## Author Comment (AC3)

"An interesting and novel approach is proposed for measuring turbine performance in time in the form of a virtual sensor signal called TPI"

➢ Thank you for your comments that are very much appreciated. We have adjusted the title and conclusion to include mention of a key contribution of the work being the introduction and use of TPI.

Updated *title*:
"Full-Scale Wind Turbine Performance Assessment using TPI: A Study of Aerodynamic Degradation and Operational Influences"

*Conclusion* key contribution highlighted:
"In response, a methodological framework was developed, integrating SCADA data with detailed O&M records to assess the impact of blade aerodynamic modifications. This approach aims to isolate the effects of these modifications amidst a multitude of factors influencing turbine performance. A key contribution of this work is the development of a controller-informed Turbine Performance Integral (TPI) method for the investigated turbine. Furthermore, STL is employed to further isolate long-term and trends and seasonal variations in performance.

The proposed methodology focuses on the isolating the individual contributions of various factors to performance deviations. Notably, the efficacy of the TPI method is demonstrated by its ability to detect distinct seasonal variations in individual turbine performance without relying on direct wind speed measurements, comparison to other turbines, or the use of combined data sources."

1. There are numerous typos and grammatical errors that need to be fixed.

➢ Thank you for pointing this out. We have carefully reviewed and corrected these in the revised manuscript to ensure clarity and precision in our presentation throughout the document.

2. Citations are mixed with the body of text, leading to misunderstanding. I suggest wrapping them in brackets.

➢ Corrected throughout the document.

3. Line 188: The range between 20% and 37% seems quite specific. How was it defined? Visually or according to a max allowed change in pitch?

➢ Text improved for clarity:

"Datasets of each turbine are loaded for the twelve year period. The integral or area under the generator speed curve between 20% and 37% of normalised power is monitored, a region empirically determined by observing linear segments of this curve - where the pitch angle is minimally active (see Figures 1 and 2). Buffers are added on each side of the range to accommodate transient behaviour."

4. Section 2.8 General comment: Brief explanations of the methods implemented, the context, and the reasons for the analyses are missing.

➢ Method section Improved:

"Statistical analysis

[revised manuscript text omitted]

5. Line 248 - "Normality Assessment": Ambiguous. More context is needed in terms of "what" and "why".

➢ Please see improved Statistical Analysis section above.

6. Line 389: How is the time window selected for the calculation of the gradient?

➢ The gradient was calculated between two points: the first two weeks prior to an event, accounting for repair time - and another three weeks after the event to capture potential trend deviations.

---

## Author Response (AR2)

Dear Peer Reviewer,

Thank you for your valuable and insightful comments on our manuscript. Your feedback has been instrumental in improving the quality and clarity of the paper.

Attached, you will find a detailed response to each of your comments, along with an updated version of the manuscript reflecting these improvements.

Thank you once again for your thorough review and constructive suggestions.

Best regards,

Tahir Malik

**Review of "Full-Scale Wind Turbine Performance Assessment using TPI: A Study of Aerodynamic Degradation and Operational Influences" by Tahir H. Malik and Christian Bak**

I clearly recommend this work to be published as the novelty of the methodology TPI is not seen before, secondly the analysis performed, and data visualization is excellent. I foresee that that the TPI could by paramount methodology to understand the change in behavior of turbines and also has the possibility to be a substitute for the old fashioned way of determining power curves (some work still needed to be done) – this not even discussed in this paper and I urge the authors to pursue this methodology – which could supersede the contractually defined power curves in the endless discussion of defining the free wind and compare to the power curve – by instead using turbine parameters as the TPI. (A possible suggestion for a new paper).

The paper has identified several contradiction mechanism that shows how degradation and upgrades of the turbines makes it very difficult to isolate the LE erosion effect on the effect of the turbines and such this is a very nice piece of work

I have the following comments for suggested need to be changed

-   Abstract Line 1-13

The findings that the change or upgrade of the controllers in the turbines the first 10 years have improved the performance of the turbines according to figure 2 needs to be included on the conclusion of the abstract more clearly.

➢ Abstract amended

-   Introduction line 15-59

The sensitive to LE erosion depends of the aero dynamic shape of the blades – here it would be nice if there was a description of this and an expected theoretical range of sensitivity due the different shapes.

➢ Thank you for your valuable feedback regarding the sensitivity of LEE to the aerodynamic shape of the blades. We appreciate the suggestion to include a description and theoretical range of sensitivity. While we recognize the importance of this topic, we have chosen to focus our current analysis on empirical data and methodologies directly related to our research objectives. We will certainly consider your recommendation for future work, where a more detailed exploration of blade shapes and their aerodynamic implications can be included. Thank you again for your input.

- Method line 60- 70

Please clarify if the 12 wind turbine are from the same wind farm and all are the same type ? also, in relation the question above in the introduction. This seem to be case from description further down ?

➢ Text amended for clarity: "*turbines within the same wind farm*"

- Line 171 -203

2.5 Turbine Performance Integral: A method for assessing wind turbine performance.

The TPI methodology is so central for the paper so I suggest that you introduce an mathematical description of the TPI in a formula at line 189 and also describe how you use it in mathematical terms

➢ Mathematical description added

- Section 3.1 is very well written and I suggest that the range where you calculate TPI is shown graphical on the figure1 and figure 2
➢ An additional Figure 1 has been added to Section 2.5 for this purpose.

3 .6 Statistical analysis this part is a really nice job and the methodologies can certainly be used in the monitoring of repairs and enhancement of the performance by OEM's

- A conclusion could be made on the blade enhancement that these should be done together with adjusting the controllers which purely belong to OEMs and therefor buying this by third party vendors is merely a vast of money and damaging for the performance.

➢ Text sharping this conclusion added

- From figure 8 and combined with figure 2 there seems to be an benefit of the software upgrade the significance test however fails – this might be due just to one outlier – if outliers was removed would this change the picture (i.e apply a filter of e.g. 3 x stdv on the TPI ) on the significance test ?

➢ Thank you for your encouragement. Indeed digging deeper into the statistical aspects is the intention of a further scientific paper.